# Canine Euthanasia’s Trend Analysis during Thirty Years (1990–2020) in Italy: A Veterinary Hospital as Case Study

**DOI:** 10.3390/vetsci11050224

**Published:** 2024-05-17

**Authors:** Annalisa Previti, Vito Biondi, Michela Pugliese, Angela Alibrandi, Agata Zirilli, Mariana Roccaro, Angelo Peli, Annamaria Passantino

**Affiliations:** 1Department of Veterinary Sciences, University of Messina, 98168 Messina, Italy; annalisa1.previti@unime.it (A.P.); vito.biondi@unime.it (V.B.); annamaria.passantino@unime.it (A.P.); 2Unit of Statistical and Mathematical Sciences, Department of Economics, University of Messina, 98122 Messina, Italy; angela.alibrandi@unime.it (A.A.); agata.zirilli@unime.it (A.Z.); 3Department for Life Quality Studies, University of Bologna, 47921 Rimini, Italy; mariana.roccaro2@unibo.it (M.R.); angelo.peli@unibo.it (A.P.)

**Keywords:** dogs, euthanasia, trend, animal welfare legislation

## Abstract

**Simple Summary:**

This study illustrates the data relating to euthanasia in a canine population during the years 1990–2020 in a small animals’ veterinary hospital. The overall period examined has been split into two terms (T1 = 1990–2004 and T2 = 2005–2020) based on Law 189/2004 coming into force, as this law made changes to the Criminal Code regarding offenses against animals and the related penalties. In comparing the significant differences of the two terms, variables such as age, breed, reproductive status, and ownership were considered. Law 189/2004 has played an important role in Italy by acknowledging animals as sentient beings and, therefore, influencing the decision-making process of euthanasia.

**Abstract:**

This study aimed to investigate changes in the number of, and reasons for, requests for dogs’ euthanasia over the last thirty years. Data (breed, age, gender, neuter status, manner, and cause of death) from dogs’ euthanasia registered between 1990 and 2020 in a small animals’ veterinary hospital were analyzed. The overall period examined has been split into two terms (T1 = 1990–2004 and T2 = 2005–2020) considering the introduction and enforcement of Law 189/2004. During the whole period examined, a significant increasing trend in euthanasia cases has been recorded (*p* = 0.027). Comparing the two terms, we observed significant differences regarding variables such as age, breed, reproductive status, and ownership. The number of euthanized dogs was significantly higher in T1 than in T2. Dogs euthanized in T2 were older than dogs in T1. A high percentage of the euthanized dogs were crossbred and stray dogs. Additionally, the number of neutered/spayed dogs was significantly higher. Regarding the cause of death, a significantly higher percentage of neoplastic processes was detected in T2. The data here reported suggest a potential influence of Law 189/2004. This law in Italy has proven to be a legal milestone that has influenced the decision-making process between euthanasia and natural death.

## 1. Introduction

A critical principle in the legal context of euthanasia is the concept of being “deemed to be in the patient’s best interest”. This signifies that the decision should be rooted in a medical expert’s judgment, the individual’s values, and the availability of treatment resources [1]. However, the Italian law known as Law 189 of 20 July 2004—which came into force to make changes to the Criminal Code about offenses against animals and the related penalties to be applied—revolutionized the treatment of animals by acknowledging them as sentient beings, shifting their status from mere property to beings capable of experiencing emotions such as pleasure and suffering [2,3]. This legal milestone, effective as of 1 August 2004, involved the amendment of the Criminal Code to include protections for animals, aligning with the acknowledgment of animals’ sentience in European legislation [1,4]. Although the circumstances under which euthanasia can be deemed necessary are not legislated, Law 189/2004 also prohibits the killing of animals if performed cruelly or “without necessity”. This lack of clarity makes it challenging to accurately determine situations of necessity, which is essential for protecting animal rights and ensuring the professional responsibility of veterinarians. Indeed, in Italy, starting in 2004, a veterinarian can face charges of killing if he/she causes the death of an animal (therefore euthanizes the animal) “without necessity” [3].

In this study, we aim to investigate clinical data collected from an anonymized small animals’ veterinary hospital to evaluate whether or not there was an increase in the prevalence of euthanasia of dogs over the last three decades.

## 2. Materials and Methods

### 2.1. Data Collection and Study Population

A retrospective analysis was conducted using anonymized clinical data retrieved from the electronic medical records database of a small animals’ veterinary hospital. The cohort considered for analysis consisted of canine patients who died from unassisted or accidental causes or were euthanized between 1 January 1990 and 31 December 2020.

Data extracted from electronic clinical records included breed, age, gender (male/female), neuter status, manner of death [euthanasia (E), unassisted (UD) or accidental (AD)], and cause of death, categorized into one of the following six physiopathological processes (PPs) [5]: infectious/inflammatory conditions, degenerative diseases, neoplastic processes, toxins, traumatic events, and other (including congenital diseases, metabolic disorders, and vascular diseases).

Assuming that, since 2005, the incidence of E has gradually leveled off due to Law 189/2004, the overall period examined has been split into two terms. Term 1 (T1) considered the data of dogs euthanized between 1990 and 2004, while Term 2 (T2) studied the data of dogs euthanized between 2005 and 2020. The terms were identified to compare data before (1990–2004) and after (2005–2020) the enforcement of the law. Although the law was passed in August 2004, 2004 was considered to be a year in which the law was not yet applied in practice.

### 2.2. Statistical Analysis

Statistical analysis was performed using the SPSS package for Windows (version 22.0, SPSS, Inc., Chicago, IL, USA).

Categorical variables [breed, gender (male/female), neutered/spayed status, ownership, manner of death, and PPs] were expressed as absolute frequency. The numerical variable of age (in months) was expressed as mean and standard deviation.

In T1 and T2, E rates were calculated as the ratio between the number of cases and the number of access/clinical records/canine patients who died.

For both terms, the tendency index was calculated on the E rates to evaluate the trend in the series. Furthermore, over the whole period, the Cox and Stuart test was applied for the trend analysis on the whole period examined (T1 + T2).

The Chi-Square test was applied to assess the existence of significant differences between the two terms, regarding categorical variables, such as breed (purebred or crossbred), sex (male/female), neutering/spaying (yes/no), and ownership (owned/stray), within the group of euthanized dogs.

A post-hoc test was used for the comparison between multiple variables.

In order to evaluate significant differences in numerical variables, such as age, the Mann–Whitney test was applied. Significance was set at *p* < 0.05.

## 3. Results

### 3.1. Study Population

A total of 80,278 canine clinical records, including those of owned and stray dogs, were examined. The veterinary hospital examined as a case study offered support to the municipality in the care of stray and free-roaming dogs diseased or injured in the absence of a health shelter. The study population consisted of n. 477 dogs who died from 1990 to 2020. At the time of death, the median age was 93.48 months. Overall, 51.8% (n = 247) of dogs were females and 48.2% (n = 230) were males, of which 14% (n = 67) were neutered/spayed. Additionally, 50.3% (n = 240) were purebred, while 49.7% (n = 237) were cross-bred. Furthermore, 81.1% (n = 387) were owned and 18.9% (n = 90) were stray. Regarding the manner of death, 72.5% (n = 346) dogs were euthanized (E), 24.4% (n = 116) died unassisted (UD), and 3.1% (n = 15) died accidentally (AD), as shown in Figure 1.

PPs included infection/inflammatory conditions (47%; n = 224), neoplastic processes (19.3%; n = 92), traumatic events (11.5%; n = 55), degenerative diseases (11.7%; n = 56), toxic disorders (5.7%; n = 27), and other conditions including congenital disorders and vascular and metabolic diseases (4.8%; n = 23) (Table 1).

Of the 477 clinical records examined, 37.3% (n = 178) of dogs were allocated to T1, while 62.7% (n = 299) were allocated to T2.

A significant increasing trend in E cases throughout the whole period (1990–2020; T1 + T2) was detected (*p* = 0.027) (Figure 2). No significant difference was identified when comparing the two terms.

From 1990 to 2020, n. 346 dogs were euthanized, of which 50.3% were euthanized in T1 and 49.7% in T2.

The median age at the time of E was 94.57 months. Out of these animals, 56.6% (n = 196) were females and 43.4% (n = 150) were males; 55.5% (n = 192) were purebred and 44.5% (n = 154) were cross-bred. Moreover, 85.5% (n = 296) were unneutered/unspayed, while 14.5% (n = 50) were neutered. Regarding ownership, 83.5% (n = 289) were owned dogs, while 16.5% (n = 57) were stray dogs. Regarding PPs, the highest number of dogs were euthanized due to the presence of infectious/inflammatory conditions (43.9%; n = 152), followed by neoplastic processes (23.7%, n = 82), degenerative diseases (14.5%, n = 50), traumatic events (7.5%, n = 26), toxins (6.1%, n = 21), and, at last, other conditions (4.3%; n = 15) (Table 2).

### 3.2. Dogs Euthanized in T1 (1990–2004)

Dogs submitted to E in T1 were n. 174. The mean age was 72.4 months (±29.5). Of these, 71.3% (n = 124) were purebred and 28,7% (n = 50) were crossbred. Regarding sex, the percentages of males and females were similar in the purebred group (72.2% vs. 70.5%) to in crossbred dogs (29.5% vs. 27.8%). No significant differences between the sexes were detected. Although the purebred dogs were older than the crossbred dogs (75.96 ± 39 vs. 69 ± 39), this difference was not significant. A total of 69.4% of dogs were neutered/spayed. No significant differences were detected regarding this variable, although the number of females spayed was significantly higher than that of males (95 vs. 79; *p* = 0.05). A total of 92.5% (n = 161) were owned, while 7.5% (n = 13) were stray dogs. No significant differences were detected in the comparison between the number of owned and stray dogs, although owned dogs were significantly older than stray dogs (76.1 ± 43.5 vs. 49.9 ± 39.9; *p* = 0.027). The majority of dogs were euthanized for infective/inflammatory conditions. Table 3 reports the percentage of dogs euthanized based on different PPs. A higher percentage of females than males were euthanized due to the presence of neoplastic processes (*p* < 0.001) (Table 3).

Table 4 reports the ages of the euthanized dogs considering the PPs. Dogs that were euthanized due to neoplastic processes were significantly older than dogs euthanized due to infectious/inflammatory conditions, toxins, traumatic events, and other causes including congenital diseases, metabolic disorders, and vascular diseases.

Dogs euthanized due to the presence of infectious/inflammatory conditions were younger than dogs euthanized due to other PPs.

The percentage of purebred dogs euthanized due to degenerative processes was higher than that of crossbred dogs (17.7% vs. 6%; *p* = 0.046). The percentage of stray dogs euthanized due to traumatic events was higher than that of owned dogs (30.8% vs. 5.6%; *p* = 0.001).

### 3.3. Dogs Euthanized in T2 (2004–2020)

There were n. 172 dogs euthanized in T2. The mean age was 100.8 months (±60.9). A total of 60.5% (n = 104) were crossbred and 39.5% (n = 68) were purebred. Regarding the crossbred dogs, 59.4% (n = 60) were females, while 62% (n = 44) were males. Among the purebred dogs, 40.6% were females (n = 41) vs. 38% (n = 27) males. Among the neutered/spayed dogs euthanized (21.5%, n = 37), 28.7% (n = 29) were females and 11.3% (n = 8) were males, while, among unneutered/unspayed dogs (78.5%, n = 135), 71.3% (n = 72) were females and 88.7% (n = 63) were males. No differences were detected regarding the variables of breed and ownership. Regarding ownership, 74.4% (n = 128) of euthanized dogs were owned, of which 75.2% (n = 76) were females and 73.2% (n = 52) were males. Among the 25.6% (n = 44) of euthanized dogs who were stray, 24.8% (n = 25) were females and 26.8% (n = 19) were males. No significant differences were identified when comparing breed and ownership. The number of females spayed (n = 101) was significantly higher (*p* = 0.006) than the number of males neutered (n. 71). Regarding the variable of age, significant differences were recorded between euthanized owned dogs and euthanized stray dogs (123.52 months vs. 89.88 months; *p* = 0.001). No significant differences were recorded regarding the number of dogs affected by various PPs (Table 5).

Although the age of dogs euthanized due to neoplastic processes was higher than that of dogs euthanized due to other PPs, this result was not statistically significant (*p* > 0.05) (Table 6). Additionally, no significant differences regarding the sex of the euthanized dogs due to PPs were recorded.

### 3.4. Distribution and Comparison of the Dogs Euthanized in the Two Periods (T1 and T2)

Table 7 summarizes data collected regarding dogs euthanized in the two terms.

No significant differences were detected in the number of euthanized dogs in the two compared terms (n. 174 in T1 vs. n. 172 in T2). Additionally, no significant differences were observed regarding sex (n = 95 females in T1 vs. n = 101 in T2; n = 79 males in T1 vs. n = 71 in T2).

A significant difference was observed regarding age. The dogs euthanized in T2 were older than the dogs euthanized in T1 (*p* = 0.004). A significant difference (*p* < 0.001) regarding the breed was also observed (Table 7). A higher percentage of crossbred dogs, 60.5% (n = 104), was detected in T2 compared to in T1, 28.7% (n = 50).

The number of neutered/spayed dogs was significantly higher (*p* < 0.001) in T2 than in T1 [21.5% (n = 37) vs. 7.5% (n = 13)] (Table 7).

A significant difference (*p* < 0.001) was identified between the two terms in the number of owned dogs euthanized. The percentage of owned dogs euthanized was significantly higher in T1 than in T2 [92.5% (n = 161) vs. 74.4% (n = 128) (*p* <0.001)], while the percentage of stray dogs euthanized was significantly higher in T2 than in T1 [25.6% (n = 447) in vs. 7.5% (n = 13) (*p* < 0.001)] (Table 7).

Regarding PPs, a significantly higher percentage of dogs with neoplastic processes present was detected in T2 than in T1 [28.5% (n = 49) vs. 19% (n = 33) (*p* = 0.038)]. Conversely, the percentage of dogs euthanized for infection/inflammatory diseases was lower in T2 than in T1 [37.2% (n = 64) vs. 50.6% (n = 88)]. No significant differences were observed when comparing other PPs.

## 4. Discussion

Before the enforcement of Law 189 on 20 July 2004, in Italy, the circumstances under which the euthanasia of small animals was justified were only partially regulated by law n. 281/1991, making it statutory that stray dogs could be euthanized only if they were considered ‘seriously or incurably ill or proven to be dangerous’ [2]. Based on the hypothesis that the enforcement of Law 189/2004 could have influenced the decision-making process surrounding euthanasia, this study has evaluated the euthanasia trend within a population of dogs examined in a small animals’ veterinary hospital over a whole period of thirty years by comparing two terms, namely, before and after the enforcement of the Italian Law 189 of 20 July 2004.

As reported by Pegram et al. [6], euthanasia has been recorded as the main cause of death in the population of dogs examined. In analyzing the trend over the last thirty years, the results have shown evidence of a substantial increase. In recent years, there has been a great focus on mortality and longevity in canine populations [5,7,8,9]. The process of making decisions regarding euthanasia can pose a moral dilemma for veterinarians and evoke emotional distress for pet owners [1,10]. Owner-related factors, such as emotional considerations, financial constraints, and time constraints in caring for a sick dog, play a significant role in influencing the decision-making process between euthanasia and natural death. The substantial increase in the rate of euthanasia herein observed may be related to a heightened awareness among pet owners of preventing and alleviating animal suffering due to their being considered sentient beings. The pivotal legislative change in Italy during the examined period was due to the enforcement of Law 189/2004, which, among other things, prohibits the killing of animals if performed cruelly or “without necessity”. This legal framework poses a challenge in determining the necessity of euthanasia for safeguarding animal rights and maintaining veterinarians’ professional accountability. The increasing significance of decision-making regarding euthanasia between veterinarians and owners highlights the need to prioritize the well-being and health of animals before resorting to euthanasia, irrespective of age. Euthanasia should be considered a tool to alleviate suffering that is unrelated to age, with a focus on minimizing distress and ensuring optimal care [6].

Our data showed that the dogs euthanized in T2, after the enforcement of Law 189/2004, were significantly older than the dogs euthanized in T1. “Old age” is considered a primary risk factor for euthanasia [11,12,13]. In modern society, the heightened focus on animals leads to a greater emphasis on their well-being, even during the geriatric phase of their lives [14,15,16]. Euthanasia is often viewed as a compassionate act aimed at alleviating unnecessary suffering in animals, regardless of their age [17]. According to Cooney and Kipperman [18], the guiding principle is that, if a life is to be ended, it should be ended at the appropriate time and for the right reasons.

On the other hand, it is plausible that age plays an important role in predicting euthanasia outcomes in stray dogs. It has been reported [17] that adult and senior stray dogs have around a four times higher likelihood of being euthanized compared to puppies. The owned dogs euthanized in T2 were older than the stray dogs euthanized in this period. This could be related to the greater longevity of owned dogs due to the greater attention paid by owners to their animals [19]. However, even though the higher age of dogs euthanized for neoplasia was not statistically significant, the difference is probably “significant” because it may relate to the increased risk of neoplasia at higher ages [20].

The reported data showed that the number of spayed bitches euthanized in T2 was higher than that of neutered males. This could be due to a greater awareness among pet owners about the risks for bitches [21]. The advantages of spaying are widely acknowledged among pet owners. Female spaying, for instance, prevents the physical manifestations of estrus and eliminates the potential risk of pyometra, a serious condition that affects over 20% of intact female dogs [22] and, in high-risk breeds, over 50% [23]. Bitches spayed before their first estrus cycle demonstrate a decreased likelihood of developing cancerous mammary tumors, with significant protective effects observed when spaying is performed before 2.5 years of age [22]. Statistics from canine cancer databases reveal that mammary tumors are prevalent, accounting for up to 76% of cancer cases in bitches, and the occurrence of these tumors decreases in dog populations with higher spaying rates [24]. While the advantages of spaying bitches are widely acknowledged, spaying can potentially increase the risk of certain cancers, musculoskeletal issues, and hormonal disorders [8]. Despite the recognized benefits of spaying, there appears to be cultural reluctance towards castrating dogs among owners in Italy [21]. It is conceivable that some owners in rural regions prefer to preserve the protective instincts of guard dogs by avoiding neutering or spaying [21].

The percentage of neoplastic processes reported was significantly higher in T2 than in T1. It is reasonable that the option of euthanasia may be reached when dogs are in pain/suffering, with a poor quality of life, and a poor prognosis [18]. The shift toward pets being considered as family members, coupled with advancements in medical practices, has led to an extended lifespan for dogs and cats [25]. This prolonged life expectancy has consequently brought about an increase in age-related pathologies, notably neoplasms [19]. Nowadays, neoplastic diseases cause approximately half of all deaths in dogs aged over ten, with around one in four dogs developing cancer at some point in their lifetime [26]. Similar to human patients, animals diagnosed with oncological conditions not only endure the localized effects of the tumor but also face systemic issues stemming from the cancer’s spread. These systemic effects significantly impact the patient’s overall well-being and quality of life [27]. The significant role that animals play in the lives of their human counterparts as cherished family members raises concerns regarding end-of-life care for dogs and cats. In the Authors’ opinion, this concern underscores the potential justification for the increased utilization of euthanasia in cases of neoplastic processes. A higher number of euthanized crossbreed dogs was observed in T2 compared with T1. This trend may be linked to the owners’ perceptions of their animals’ health status [28], which can be affected by the dogs’ breed [18]. Previous studies have indicated that the strength of the bond between a dog and its owner influences the health-seeking behaviors of dog owners [29]. For instance, owners with strong bonds tend to seek higher levels of veterinary care and are more inclined to follow veterinary recommendations, regardless of cost. However, studies focusing on owners of particular breeds, such as brachycephalic dogs, have identified discrepancies in their perceptions of dog health compared to those of veterinary professionals [30]. These differences may disrupt the tendency of some owners to adhere to veterinary advice: this is a behavior commonly observed in dog owners in general. These owners often normalize poor health conditions within their breed [30]. Despite recognizing signs of respiratory issues in their dogs, they may not acknowledge these as serious concerns but rather view them as typical characteristics of the breed [31]. This normalization extends beyond respiratory problems to include issues like abnormal sleeping patterns, thermoregulation difficulties, and eating habits that are considered normal in some breeds. As a result, owners may only recognize a problem in their dog when it reaches a critical level of severity [32]. These normalization and thresholding phenomena can influence euthanasia decision-making, as owners may fail to perceive their dog as unwell and may not believe that their dog’s quality of life is significantly compromised to warrant euthanasia. Consequently, severely affected dogs are more likely to experience a natural death without assistance [18].

On the other hand, in Italy, the euthanasia of stray dogs is permissible only in cases of an incurable condition, such as terminal illness, or demonstrated aggressivity [33]. In other states, such as California, euthanasia may also be employed to manage shelter overcrowding and mitigate the spread of infectious diseases [34,35]. Various animal welfare organizations have attempted to avoid euthanasia in these circumstances, prompting many shelters to adopt no-kill policies for adoptable animals [36,37]. While this approach yields benefits for the animals, it also entails drawbacks, such as heightened costs and increased space requirements. Consequently, countries like the United States continue to utilize euthanasia practices, despite the situation regarding animal euthanasia in North American shelters being identified as a long-lasting impediment [34], and a substantial decline in euthanasia rates in shelters has been reported [38,39,40]. According to Rogelberg et al. (2007) [41], a reduction in or cessation of the euthanasia of healthy animals is considered a positive goal for animal shelters.

The higher number of euthanasias of stray dogs here reported may be linked to heightened regard for dogs, irrespective of their ownership status, and/or an attempt to alleviate their pain/suffering, also in consideration of Law 189/2004.

Upholding animal welfare and dignity as fundamental principles in veterinary practice necessitates a deeper comprehension of disease progression and ethical animal pain management. These insights have the potential to shape future research endeavors, particularly qualitative studies aimed at evaluating discussions on the quality of life, developing criteria for assessing the suitability of euthanasia for specific conditions and examining the impact of euthanasia decision-making on both pet owners and veterinary professionals. Considering the legal aspects of euthanasia, these are aimed at preventing and reducing the suffering of animals [42], as well as preserving the quality of death. Analyzing the E trend in the thirty years examined, it is possible to assert that the enforcement of Law 189/2024 has influenced the decision-making process of euthanasia. In fact, from the assessment of data before and after the enforcement of the Law, it is possible to evaluate how it has influenced the number of euthanasias performed, as well as correlated variables such as the age of dogs, the breed, the ownership, and the reproductive status.

## 5. Conclusions

Law 189/2004 in Italy has shown to be a legal milestone that influenced the decision-making process surrounding euthanasia and natural death. Supporting animal welfare and dignity are essential values in veterinary practice requiring a greater comprehension of illness evolution and pain management.

## Figures and Tables

**Figure 1 vetsci-11-00224-f001:**
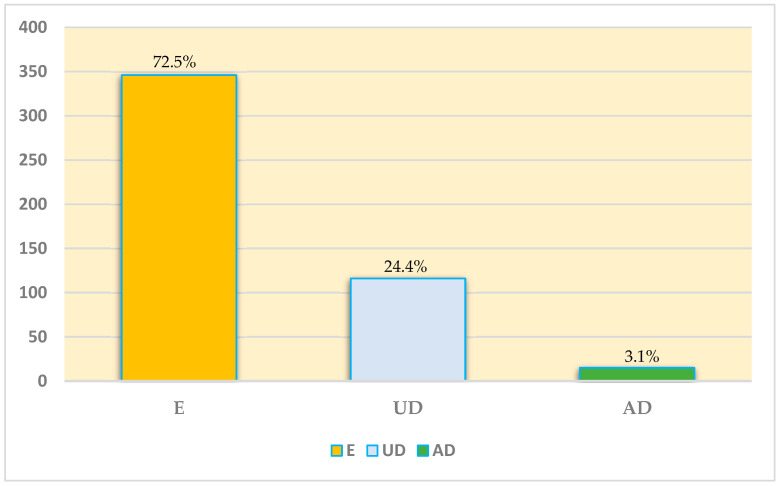
The number of dogs that died from 1990 to 2020 (T1 + T2) distributed in relation to the manner of death. E = euthanasia; UD = unassisted death; AD = accidental death.

**Figure 2 vetsci-11-00224-f002:**
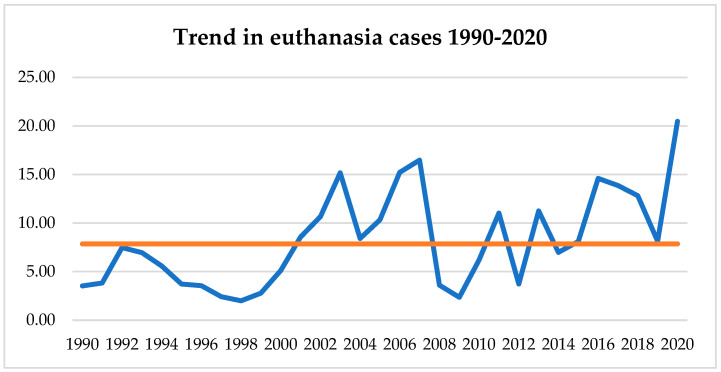
Trend in euthanasia cases throughout whole period examined (1990–2020; T1 + T2) (*p* = 0.027). The orange line refers to the mean of the number of cases in each year.

**Table 1 vetsci-11-00224-t001:** Variables identified in the population of dogs examined (n. 477) who died from 1990 to 2020 (T1 + T2). PPs = pathological processes; E = euthanasia; UD = unassisted death; AD = accidental death.

Variables		No.	%
Age	M	230	48.2
	F	247	51.8
Breed	Purebred	240	50.3
	Cross-bred	237	49.7
Status	Neutered/Spayed	67	14
	Unneutered/Unspayed	410	86
Ownership	Owned	387	81.1
	Stray	90	18.9
Term	T1	178	37.7
	T2	299	62.7
PPs	Infective/Inflammatory conditions	224	47
	Neoplastic processes	92	19.3
	Degenerative diseases	56	11.7
	Traumatic events	55	11.5
	Toxic	27	5.7
	Others	23	4.8
Death	E	346	72.5
	UD	116	24.3
	AD	15	3.1

**Table 2 vetsci-11-00224-t002:** Variables identified in the population of 346 dogs euthanized from 1990 to 2020. PPs = pathological processes.

Variable		No.	%
Age	M	196	56.6
	F	150	43.4
Breed	Purebred	192	55.5
	Crossbred	154	44.5
Neutered	Neutered/Spayed	50	14.5
	Unneutered/Unspayed	296	85.5
Ownership	Owned	289	83.5
	Stray	57	16.5
Term	T1	174	50.3
	T2	172	49.7
PPs	Infective/Inflammatory conditions	152	43.9
	Neoplastic processes	82	23.7
	Degenerative diseases	50	14.5
	Traumatic events	26	7.5
	Toxic	21	6.1
	Others	15	4.3

**Table 3 vetsci-11-00224-t003:** Distribution of PPs considering the variable sex in T1 (1990–2004).

PPs	Sex		Tot.	%	*p*-Value
	F	M			
Infective/Inflammatory conditions	42	46	88	50.6	Ns
Neoplastic processes	29	4	33	19	<0.001
Traumatic events	5	8	13	7.5	Ns
Degenerative diseases	12	13	25	14.4	Ns
Toxic	6	3	9	5.2	Ns
Others	1	5	6	3.4	Ns

PPs = pathological processes; Ns = not significant; F = females; M = males.

**Table 4 vetsci-11-00224-t004:** Age of dogs euthanized distributed considering the PPs in T1 (1990–2004).

PPs	Age (Months)	
	Mean	SD
Inflammatory/Infectious conditions	59.9 ^AB^	42.0
Neoplastic processes	111.4 ^C^	38.8
Degenerative diseases	92.6 ^B^	39.6
Traumatic events	51.9 ^C^	15.6
Toxic	64.6 ^C^	18.6
Others	54 ^C^	22.5

Capital letters indicate the significance of the column based on the variable age for a specific pathological process (PPs) compared with other PPs. SD = standard deviation.

**Table 5 vetsci-11-00224-t005:** Distribution of PPs in T2 (2005–2020).

PPs	Tot.	%	*p*-Value
Infective/Inflammatory conditions	133	44.5	Ns
Neoplastic processes	59	19.7	Ns
Traumatic events	42	14	Ns
Degenerative diseases	31	10.4	Ns
Toxic	17	5.7	Ns
Others	17	5.7	Ns

PPs = pathological processes; Ns = not significant.

**Table 6 vetsci-11-00224-t006:** Age of dogs euthanized distributed considering the PPs in T2 (2004–2020).

PPs	Age (Months)	
	Mean	SD
Inflammatory/Infectious conditions	111.8	61.5
Neoplastic processes	134.2	53.4
Traumatic events	103.6	60.5
Degenerative diseases	94.7	55.9
Toxic	106	73.9
Others	54.5	60.2

SD = standard deviation.

**Table 7 vetsci-11-00224-t007:** Comparison of variables of dogs euthanized in two terms (T1 and T2).

Variables		T1		T2		*p*-Value
		No.	%	No.	%	
Sex	M	79	45.4	71	41.3	Ns
	F	95	54.6	101	58.7	
Breed	Purebred	124	71.3	68	39.5	<0.001
	Cross-bred	50	28.7	104	60.5	
Status	Neutered/Spayed	13	7.5	37	21.5	<0.001
	Unneutered/Unspayed	161	92.5	135	78.5	
Ownership	Owned	161	92.5	128	74.4	<0.001
	Stray	13	7.5	44	25.6	
PPs	Inflammatory/Infectious conditions	91	51.1	133	44.5	Ns
	Neoplastic processes	33	18.5	59	19.7	Ns
	Traumatic events	13	7.3	42	14	Ns
	Degenerative diseases	25	14	31	10.4	Ns
	Toxic	10	5.6	17	5.7	Ns
	Others	6	3.4	17	5.7	Ns

PPs = pathological processes; Ns = not significant.

## Data Availability

For other information contact the corresponding author.

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
