# Peer review of "Canine Euthanasia’s Trend Analysis during Thirty Years (1990–2020) in Italy: A Veterinary Hospital as Case Study"

_vetsci, 2024, doi:10.3390/vetsci11050224_

Round 1
Reviewer 1 Report
Comments and Suggestions for Authors
Lines 40-43. The emotional bond between dog owners and their beloved pets is continuously 40 evolving and strengthening (1), influencing the quality of care that the animals receive 41 and end-of-life decisions for these animals. This shift in focus from the quantity to the 42 quality of an animal’s life underscores the ethical significance placed on animal welfare.
The paper begins with several assumptions that are not actually well-supported by the literature. First, the authors take as given that “the human animal bond has strengthened over the past 40 years.” Although this trope appears frequently in both popular and scholarly writing, it is not actually based on good evidence and is highly debatable. The authors further seem to suggest that a “strengthening of the human animal bond” is causally related to a shift in care practices and attitudes described here as a shift in focus from quantity to quality of life. Again, although this claim finds its way into the literature over and over, it has no evidentiary grounding. Next, the authors say that “nowadays companion animal owners see their pets as family.” Once again: this “pets are now considered family” claim crops up all over the place in the literature, but it is not based on solid research. I would recommend simply cutting the whole first paragraph. In fact, I think the paper would be strengthened if the authors began at line 69, getting straight to the point of their research.
Line 85. This shift in attitude has led to a substantial improvement in quality of life, and, consequently, in animal welfare standards (18).
I don’t see how this claim is substantiated. Broom’s article (citation 18) was not focused on the 2004 change in Italian law. Perhaps the problem here is how the sentence is phrased. Would recommend cutting lines 85-86.
Lines 87-90. Furthermore, there is a growing interest in enhancing veterinary practices to effectively meet the evolving needs of today’s pet owners and ensure both animal welfare and 88 better care (19). In fact, improved veterinary care and quality of life can increase the longevity of dogs decreasing the prevalence of diseases (20).
Again, broad and unsubstantiated claims, and not really relevant to the authors’ research. Recommend simply deleting.
I would suggest using the word “euthanasia” in the Discussion section, rather than simply using “E.”
Line 355. Note that California is a state, not a country.
Comments on the Quality of English LanguageJournal editors should go through carefully and make suggestions to the authors.
Author Response
Dear Reviewer,
Thank you very much for your time and all your comments.
We thank you for your precise and thoughtful comments and constructive criticism, which has led to a better manuscript.
We revised the manuscript concerning the suggestions and more detailed answers are given below.
The changes made in the manuscript to address comments are written in red.
R. Lines 40-43. The emotional bond between dog owners and their beloved pets is continuously 40 evolving and strengthening (1), influencing the quality of care that the animals receive 41 and end-of-life decisions for these animals. This shift in focus from the quantity to the 42 quality of an animal’s life underscores the ethical significance placed on animal welfare.
The paper begins with several assumptions that are not actually well-supported by the literature. First, the authors take as given that “the human animal bond has strengthened over the past 40 years.” Although this trope appears frequently in both popular and scholarly writing, it is not actually based on good evidence and is highly debatable. The authors further seem to suggest that a “strengthening of the human animal bond” is causally related to a shift in care practices and attitudes described here as a shift in focus from quantity to quality of life. Again, although this claim finds its way into the literature over and over, it has no evidentiary grounding. Next, the authors say that “nowadays companion animal owners see their pets as family.” Once again: this “pets are now considered family” claim crops up all over the place in the literature, but it is not based on solid research. I would recommend simply cutting the whole first paragraph. In fact, I think the paper would be strengthened if the authors began at line 69, getting straight to the point of their research.
A. The paragraph has been modified as suggest.
R. Line 85. This shift in attitude has led to a substantial improvement in quality of life, and, consequently, in animal welfare standards (18).I don’t see how this claim is substantiated. Broom’s article (citation 18) was not focused on the 2004 change in Italian law. Perhaps the problem here is how the sentence is phrased. Would recommend cutting lines 85-86.
A. Done.
R. Lines 87-90. Furthermore, there is a growing interest in enhancing veterinary practices to effectively meet the evolving needs of today’s pet owners and ensure both animal welfare and 88 better care (19). In fact, improved veterinary care and quality of life can increase the longevity of dogs decreasing the prevalence of diseases (20). Again, broad and unsubstantiated claims, and not really relevant to the authors’ research. Recommend simply deleting.
A. Done.
R. I would suggest using the word “euthanasia” in the Discussion section, rather than simply using “E.
A. Done.
R. Line 355. Note that California is a state, not a country.
A. Done.

Reviewer 2 Report
Comments and Suggestions for Authors
It would be helpful to have some sense of the number of dogs seen as patients by the veterinary practice described in the manuscript annually and the percentage of those dogs being euthanized. I would also suggest explaining how "strays" are brought to the practice for treatment/euthanasia. The actual N for the sample is rather small. 346 animals being euthanized over 30 years amounts to an average of 11 dogs being euthanized per year. Dividing these numbers into several categories leaves the study with very small samples to compare - hence the lack of statistical significance. I am not sure why the deaths and accidental deaths are included in the analysis.
The No Kill and Open Access situation in North American shelters is a long-standing complication as regards animal euthanasia in North America. There have been substantial declines in euthanasia rates in the USA (over 90% since 1973). The debate over sterilization and its impact on individual dog longevity is a recent phenomenon in the USA. The benefits of sterilization are still being debated and the issue is far from resolved in North America.

The English is not at all bad but the manuscript could do with light copy editing to make the sense flow more smoothly. For example, lines 164 and 165 should read "From 1990 to 2020, 346 dogs were euthanized, of which 50.3% were euthanized in T1 and 49.7% in T2."
More could be made of the differences in the need for euthanasia of strays versus owned dogs as a result of accidents. Even though the higher age of dogs euthanized for neoplasia was not statistically significant, the difference is probably "significant" because of the increased risk of neoplasia at higher ages. Perhaps provide some numerical description of increasing neoplasia risk with age.
There are some additional references attached regarding euthanasia and no-kill that the authors might find helpful for context.
Author Response
Dear Reviewer,
Thank you very much for your time and all your comments.
We thank you for your precise and thoughtful comments and constructive criticism, which has led to a better manuscript.
We revised the manuscript concerning the suggestions and more detailed answers are given below.
The changes made in the manuscript to address comments are written in red.
R. It would be helpful to have some sense of the number of dogs seen as patients by the veterinary practice described in the manuscript annually and the percentage of those dogs being euthanized. I would also suggest explaining how "strays" are brought to the practice for treatment/euthanasia. The actual N for the sample is rather small. 346 animals being euthanized over 30 years amounts to an average of 11 dogs being euthanized per year. Dividing these numbers into several categories leaves the study with very small samples to compare - hence the lack of statistical significance. I am not sure why the deaths and accidental deaths are included in the analysis.
A. The number of dogs seen as patients by the veterinary practice annually and the percentage of those dogs being euthanized have been included. The veterinary hospital examined as a case study offered support at the municipality in caring for stray and free-roaming dogs diseased or injured, in the absence of a health shelter.
R. The No Kill and Open Access situation in North American shelters is a long-standing complication as regards animal euthanasia in North America. There have been substantial declines in euthanasia rates in the USA (over 90% since 1973). The debate over sterilization and its impact on individual dog longevity is a recent phenomenon in the USA. The benefits of sterilization are still being debated and the issue is far from resolved in North America.
A. The situation of the No Kill and Open Access situation in North American shelters is better described, as suggested.
R. Comments on the Quality of English Language -The English is not at all bad but the manuscript could do with light copy editing to make the sense flow more smoothly. For example, lines 164 and 165 should read "From 1990 to 2020, 346 dogs were euthanized, of which 50.3% were euthanized in T1 and 49.7% in T2.
A. Done.
More could be made of the differences in the need for euthanasia of strays versus owned dogs as a result of accidents. Even though the higher age of dogs euthanized for neoplasia was not statistically significant, the difference is probably "significant" because of the increased risk of neoplasia at higher ages. Perhaps provide some numerical description of increasing neoplasia risk with age.
A. Done.
R. There are some additional references attached regarding euthanasia and no-kill that the authors might find helpful for context.
A. Thank you for your support. The references list provided has been used to improve the MS.

Reviewer 3 Report
Comments and Suggestions for Authors
A timely, interesting and useful paper, well referenced, clearly presented and argued, suitable for publication in it's present form; however there were a couple of very minor issues of English that authors may benefit from re-writing.
Lines 301-304 The didn't quite make sense in English, although I understood the meaning.
Line 357, should the word 'such' be inserted between 'correlated' and 'as'?
Author Response
Dear Reviewer,
Thank you very much for your time and all your comments.
We thank you for your precise and thoughtful comments and constructive criticism, which has led to a better manuscript.
We revised the manuscript concerning the suggestions and more detailed answers are given below.
The changes made in the manuscript to address comments are written in red.
A timely, interesting and useful paper, well referenced, clearly presented and argued, suitable for publication in it's present form; however there were a couple of very minor issues of English that authors may benefit from re-writing.
R. Lines 301-304 The didn't quite make sense in English, although I understood the meaning.
A. The quality of the sentence has been improved.
R. Line 377, should the word 'such' be inserted between 'correlated' and 'as'?A. A. The sentence has been modified as suggested.
